

# Comparative transcriptome analysis identified candidate genes associated with kernel row number in maize

Shukai Wang, Yancui Wang, Xitong Xu, Dusheng Lu, Baokun Li, Yuxin Zhao, Senan Cheng, Zhenhong Li and Cuixia Chen

College of Agronomy, Shandong Agricultural University, Taian, Shandong, China

## ABSTRACT

Kernel row number (KRN) is a crucial trait in maize that has a high impact on yield. However, KRN is a typical quantitative trait with only a few genes being verified so far. Here, two maize inbred lines with contrasting KRN were used to perform transcriptome analysis at five early ear developmental stages. Pairwise differential gene expression analyses were performed, and a total of 11,897 line-specific differentially expressed genes (DEGs) were detected between the two lines across the five development stages. Clustering analysis of line-specific DEGs revealed that the trends of gene expression changed significantly in the five stages, thus the five stages were further divided into two development phases: Phase I (V6-V8) and Phase II (V9-V10). Gene ontology enrichment analysis revealed that different transcriptional pathways were activated in the two phases. DEGs in Phase I were significantly enriched in morphogenesis and differentiation processes and hormone regulation. Of the 5,850 line-specific DEGs in Phase I, 2,132 genes were in known quantitative trait loci (QTLs) or flanking regions of quantitative trait nucleotides (QTNs), of which 92 were repeatedly detected in QTLs where QTNs also exist. The 92 high-probability candidate genes included development-related transcription factors (SBP-box and AP2/EREBP TFs) as well as genes involved in hormone homeostasis and signaling. Our study provides genetic resources for the elucidation of the molecular mechanisms of KRN development and reference for the cloning of candidate genes.

# INTRODUCTION

Maize (*Zea mays* L.) is a staple crop for food, feed, and energy, playing a crucial role in global food security and agricultural economics. With the growing global demand for maize, increasing the yield remains a primary goal in breeding programs (*Erenstein et al., 2022*). Kernel row number (KRN) is a critical determinant of maize yield and is a complex quantitative trait controlled by genetic factors with high heritability (*Liu et al., 2015c*; *Wu et al., 2020*). Therefore, identifying genes associated with KRN has been an effective strategy for increasing the total kernel number and overall yield.

Maize ears originate from the leaf axillary meristem (AM) in the leaf axils. Upon transitioning from the vegetative growth to the reproductive phase, the AM develops into

Corresponding author
Cuixia Chen, cxchen@sdau.edu.cn

the inflorescence meristem (IM). The indeterminate IM continually produces determinate spikelet pair meristems (SPMs), which further differentiate into two spikelet meristems (SMs). Subsequently, each SM is divided into two floret meristems (FMs) (*Pautler et al., 2013*; *Tanaka et al., 2013*). The quantities of spikelets and florets directly determine KRN, which ultimately influences grain yield (*Pan et al., 2015*; *Hu et al., 2022*). Additionally, previous studies have identified complex regulatory networks responsible for determining the number of SPMs in maize ear, which involves multiple genes and pathways, including transcription factors (TFs), plant hormones, and peptides (*Zhang et al., 2023*). Although weak alleles of fasciated ears, such as *thick tassel dwarf1* (*td1*), *fasciated ear2* (*fea2*), and *fasciated ear3* (*fea3*), could enhance hybrid maize yield traits by increasing KRN (*Liu et al., 2021*; *Bommert et al., 2005*; *Bommert, Nagasawa & Jackson, 2013*; *Je et al., 2016*), these strong loss-of-function mutants with pleiotropic or extreme phenotypes are unsuitable for direct utilization. Consequently, further evidence is needed to predict and validate causal genes underlying quantitative trait loci (QTLs), and the favorable alleles in natural populations can be used as genetic markers and potential targets for crop molecular genetic improvement.

QTL mapping and genome-wide association study (GWAS) are the primary approaches for dissecting complex quantitative traits (*Liang et al., 2021*). However, due to limitations in population size and effective recombination, QTL mapping typically results in large confidence intervals, spanning 10-30 cM and containing hundreds of candidate genes (*Salvi & Tuberosa, 2005*). Association analysis is based on linkage disequilibrium (LD), and by utilizing ancient recombination events, it correlates markers with phenotypes in natural populations at a relatively high resolution (*Flint-Garcia, Thornsberry & ESt, 2003*; *Myles et al., 2009*). Typically, genes within the LD blocks surrounding quantitative trait nucleotides (quantitative trait nucleotides (QTNs), significant SNPs associated with traits) are considered as initial candidate genes (*Ferguson et al., 2021*; *Kadam et al., 2017*). Previous studies have identified numerous QTLs and QTNs for KRN through QTL mapping and GWAS. However, only a limited number of QTLs have been cloned successfully, such as *KRN1* (*Wang et al., 2019*), *KRN2* (*Chen et al., 2022*), *KRN4* (*Liu et al., 2015c*; *Du et al., 2020*), and *qKRN5b* (*Shen et al., 2019*; *Shen et al., 2024*), *qKRN5.04* (*An et al., 2022*). Currently, the majority of QTLs are mapped in broad and imprecise genomic regions that contain multiple genes. For instance, although *qKRN8* has been fine-mapped to a 520-kb interval, this interval still contains six putative candidate genes (*Han et al., 2020*).

With the advancement of sequencing technology and the continuous improvement of the maize genome assembly, transcriptome sequencing enables the rapid identification of differentially expressed genes (DEGs) within major QTLs for complex traits of interest. It has been proposed as a crucial tool for validating candidate genes (*Wang et al., 2023*). Although transcriptome studies have generated a substantial amount of gene expression data concerning various aspects of ear development in maize at different developmental stages and at single-cell levels (*Liu et al., 2015a*; *Xu et al., 2021*; *Shen et al., 2023*), there is a notable gap in research focusing on comparative transcriptome profiles of inbred lines with distant KRN at various developmental stages. In this study, we performed a high temporal resolution transcriptome analysis during the early developmental stages

of ears in two maize inbred lines that show distant KRN traits. Through comparative transcriptome analysis, line-specific (between inbred lines) and development-specific (between development stages) DEGs were identified. Furthermore, we concentrate on integrating these line-specific DEGs from the early stages of ear development into previously reported KRN-related QTLs and QTN regions. Our findings are expected to contribute to exploring the critical developmental phases associated with KRN formation and facilitating the prediction of candidate genes for KRN, thereby enhancing our understanding of the genetic architecture and molecular mechanisms involved in maize ear development.

## MATERIALS & METHODS

### Plant materials and sample collection

Two maize inbred lines with distinct phenotypes in KRN, PHG35 and Dan598, were planted in the experimental field of Shandong Agricultural University (Taian, China). Dan598, an elite line and a parent of widely used commercial hybrids in China, originated from the Lüda Honggu mixed population and introduced hybrid varieties PN78599. The Expired Plant Variety Protection (ex-PVP) germplasm PHG35, derived from G3BD2 and H7FS6, was obtained from the North Central Regional Plant Introduction Station (USDA-ARS, Ames, IA, United States). The trial plants were planted in four-row plots at a density of 66,700 plants per hectare. Row spacing and plant spacing were about 0.6 and 0.25 m, respectively. The management during the entire growing period is the same as local field management practices. At least 30 plants were used for the measurement of KRN trait in each line. For RNA-Seq, young ear tissues from both inbred lines were collected at development stages V6 through V10 with three biological replications at each stage. At the V6 and V7 stages, 25–30 individuals were pooled per biological replicate; at the V8, V9, and V10 stages, 15–20 individuals were pooled for each replicate. The samples were immediately flash-frozen in liquid nitrogen and then stored at −80 °C for subsequent analysis.

### Library construction, quality control, and read alignment

RNA isolation, cDNA library construction, and RNA sequencing were performed at Novogene Bioinformatics Technology (Beijing, China). Briefly, total RNA was extracted using a TRIzol reagent. The quality of extracted RNA was assessed using the NanoDrop ND-2000 spectrophotometer and an Agilent 2100 Bioanalyzer. After the RNA sample was qualified, mRNA was enriched using magnetic beads with Oligo(dT). Then, mRNA was fragmented and converted into single-stranded cDNA with random hexamers. Subsequently, the double-stranded cDNA was synthesized and purified with AMPure XP beads. The purified double-stranded cDNA underwent end repair, A-tailing, and adapter ligation, followed by size selection using AMPure XP beads. Finally, PCR amplification was performed, and the PCR products were purified using AMPure XP beads to obtain the final library. After library construction, a preliminary quantification was performed using Qubit 2.0, and the library was diluted. Then, the insert size of the library was detected using Agilent 2100, and accurate concentration was quantified using the Q-PCR method. All 30 cDNA libraries were sequenced on an Illumina NovaSeq 6,000 platform

with a paired-end 150 bp sequencing strategy. The raw reads underwent initial filtering and quality control processes to generate clean reads using fastp (*Chen et al., 2018*). The filtered high-quality clean reads of each sample were aligned to the B73 reference genome (https://download.maizegdb.org/Zm-B73-REFERENCE-GRAMENE-4.0) using HISAT2 version 2.2.1 (*Kim et al., 2019*). Alignments in the SAM format were converted into BAM format using SAMtools version 1.6 (*Li et al., 2009*). Transcript abundance was estimated as Fragments per kilobase of transcript per million mapped reads (FPKM) and read counts using StringTie version 2.2.1 (*Pertea et al., 2016*). The statistical power of this experimental design was calculated using RNASeqPower (*Hart et al., 2013*).

## Global transcriptome analyses and differential gene expression analysis

To assess and explore the global transcriptomic profiles, Principal component analysis (PCA) and Pearson correlation among the three biological replicates were performed (*Hou et al., 2024*; *Zheng et al., 2021*). Pairwise differential gene expression analyses were conducted by the DESeq2 package (version 1.41.12) based on read counts (*Love, Huber & Anders, 2014*). Genes with foldchange $\geq 2$ and $p$-adjusted $< 0.05$ were considered differentially expressed genes (DEGs). DEGs identified by comparing the two inbred lines at the same developmental stages were termed line-specific DEGs, and DEGs identified between two consecutive developmental stages in the same line were termed development-specific DEGs. The gene expression patterns of DEGs were clustered by the Mfuzz package of R (*Kumar & Futschik, 2007*). Gene Ontology (GO) classification was performed using the clusterProfiler package (*Wu et al., 2021*).

## Prediction of candidate genes underlying known QTLs and QTNs

To further delineate candidate genes underlying QTLs and QTNs, we compiled KRN-related QTLs (*Yan et al., 2006*; *Ma et al., 2007*; *Guo et al., 2011*; *Lu et al., 2011*; *Cai et al., 2014*; *Liu et al., 2015b*; *Yang et al., 2015*; *Zhang et al., 2017*; *Fei et al., 2022*) and QTNs (*Liu et al., 2015b*; *Zhang et al., 2017*; *Li et al., 2018*; *An et al., 2020*; *Liu et al., 2020*; *Zhang et al., 2020*; *Zhang et al., 2022*) from previous studies. Given that different reference genomes were used in these studies, the BLAST search was performed for the markers of these QTLs and QTNs against the B73 RefGen_v4 genome in maizeGDB. QTLs that overlapped across different studies ($n \geq 2$) were merged into larger QTL intervals and were defined as QTL hotspots in this study. Subsequently, we mapped the line-specific DEGs in Phase I to QTL hotspots and the 200 kb genomic region surrounding a significant QTN (100 kb each upstream and downstream), and the genes in these regions were regarded as KRN-related candidate genes for further analysis. Transcription factor databases were obtained from PlantPAN 4.0 (http://plantpan.itps.ncku.edu.tw/).

## Expression validation by quantitative real-time PCR (qRT-PCR)

A random sample from each of the three biological replicates at each stage of two inbred lines was used for reverse transcription and qRT-PCR analysis. Total RNA was extracted using the TRIzol method, and cDNA was synthesized using the Vazyme Reverse Transcription Kit R223. qRT-PCR was performed on an Applied Biosystems PCR instrument VIIA7 with

the Ultra SYBR Mixture (Cwbio, Taizhou, China), following the manufacturer's protocols. The 2-$\Delta\Delta$Ct method was employed to evaluate the relative abundance of the candidate gene, with three biological replicates. Expression levels were normalized to those of the endogenous control gene, CULLIN. The primer sequences can be found in Table S1.

## RESULTS

### Transcriptome sequencing of young ears in the two maize inbred lines

In this study, the average KRN for inbred line PHG35 was $20.5 \pm 1.3$ and was $12.1 \pm 1.1$ for inbred line Dan598 (Fig. 1). To explore transcriptional differences between the two maize inbred lines that have distinct KRN phenotypes, young ear tissues were collected from both inbred lines at development stages V6 through V10 with three biological replications each stage. A total of 30 cDNA libraries were constructed for sequencing. With a sample size of three, the average statistical power of this experimental design among different groups was 0.848. A summary of the sequencing data is shown in Table S2. An average of 45.2 million raw reads were generated for each library, and a total of 1,344.3 M high-quality clean reads were obtained after filtering (ranging from 40.13 M to 50.91 M per library; Q30 was 92.8% to 94.2%). The clean reads of each sample were then mapped to the maize B73 RefGen_v4 reference genome. The average mapping ratio was 89.40% (ranging from 87.99% to 90.96%) and the uniquely mapped reads ranged from 73.15% to 81.91%.

### Global comparison of transcriptome among the samples

Pearson correlation analysis and principal component analysis (PCA) were performed on the 30 ear samples across various ear developmental stages in PHG35 and Dan598. Correlation coefficients were higher within each inbred line compared with that for different developmental stages (Fig. 2A). The two inbred lines were clearly distinct in PCA (Fig. 2B), suggesting that the two lines with distinct KRN had notably different transcriptomic profiles. Furthermore, the three early developmental stages (V6 to V8) were distinguishable from the two later stages (V9 and V10), suggesting that different transcriptional pathways could be activated at the different ear developmental stages. However, the sample D_V10_1 exhibited separation in Fig. 2B. Consequently, to ensure adequate data quality and reproducibility among replicates, we elected to exclude the sample D_V10_1. The Spearman's rank correlation coefficients (SCC) between D_V10_2 and D_V10_3 were 0.97 (Fig. S1B), revealing sufficient data quality and reproducibility between the replicates. The average FPKM value of the biological replicate was used to determine the expression level for each stage in both lines. We found that 21,940, 21,558, 21,386, 21,804, and 22,258 genes were expressed (FPKM $\geq$ 1) at the V6, V7, V8, V9, and V10 stages, respectively, in PHG35; similarly, 21,436, 21,500, 21,417, 21,999 and 22,492 genes were found expressed at the corresponding stages in Dan598 (Fig. 2C). On average, 53.1% of the detected genes had FPKM < 1, and were considered not expressed; 22.3% had low expression ($1 \leq$ FPKM < 10), 22.6% were moderately expressed ($10 \leq$ FPKM < 100) and 2.0% were highly expressed (FPKM $\geq$ 100), without apparent difference in gene expression abundance distribution among different samples (Fig. 2D). The overall

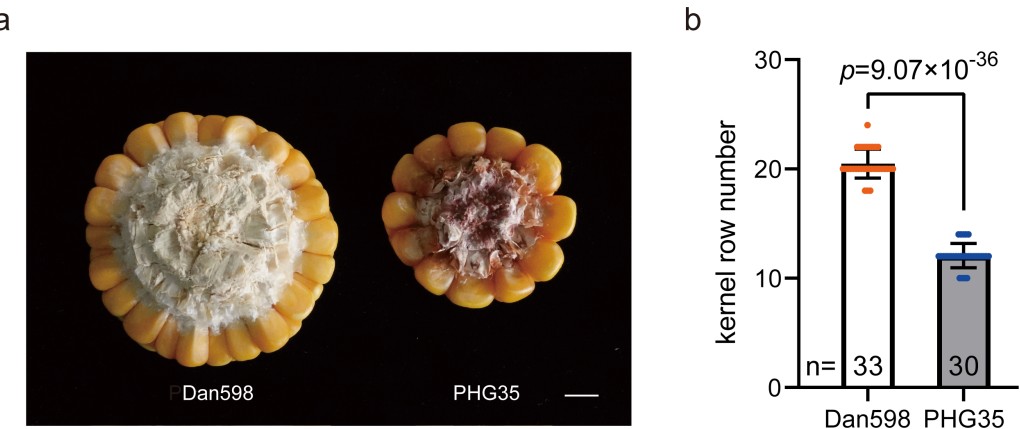

**Figure 1** **KRN phenotype of the two maize inbred lines.** (A) Phenotypic demonstration of KRN in Dan598 (left) and PHG35 (right). Scale bars, one cm. (B) KRN shows a significant difference between Dan598 and PHG35. The data is presented as means ± SD. The *p*-value is estimated by the two-tailed *t*-test. *n*, sample size.

transcriptome dynamics of maize ear development in the two maize inbred lines are depicted in Fig. 2E.

## Exploration of line-specific and development-specific differentially expressed genes

To identify line-specific DEGs, we conducted pairwise differential gene expression analyses between the PHG35 and Dan598 samples at each developmental stage. There were 3,497 DEGs (1,514 up-regulated and 1,983 down-regulated) at V6, 3,775 DEGs (2,501 up-regulated and 1,724 down-regulated) at V7, 3,508 DEGs (1,811 up-regulated and 1,697 down-regulated) at V8, 6,689 DEGs (3,651 up-regulated and 3,038 down-regulated) at V9, and 7,908 DEGs (4,244 up-regulated and 3664 down-regulated) at V10 detected in Dan598 when compared with PHG35 (Fig. 3A and Data S2). Taking all developmental stages together, we identified 11,897 DEGs between the two inbred lines, demonstrating immense differences in their transcriptomic profiles. Among these, V6-V8 stages contained 5,850 line-specific DEGs and V9-V10 stages contained 9,803 line-specific DEGs (Fig. 3B). Furthermore, we detected DEGs that were specifically expressed in one or more stages (Fig. S2). There were 454, 418, 310, 1,263, and 2,240 DEGs that were uniquely expressed in the V6, V7, V8, V9, and V10 stages, respectively. A unique set of 251 DEGs was common to both the V6 and V7 stages, including *fasciated ear3* (*fea3*). Similarly, 226 DEGs were uniquely expressed in both the V7 and V8 stages, such as *terminal ear1* (*te1*), *barren stalk1* (*ba1*), and *barren stalk fastigiate1* (*baf1*). Additionally, 353 common DEGs were specifically expressed from V6 to V8, such as *barren inflorescence2* (*bif2*). The specific expression of 998 common DEGs spanned from V6 to V10, with examples including *unbranched2* (*ub2*) and *barren inflorescence4* (*bif4*). The genes *ba1*, *baf1*, *bif2*, *ub2*, and *bif4* are involved in the initiation of lateral organs and axillary meristems (*Chuck et al., 2014*; *Gallavotti et al., 2011*; *Gallavotti et al., 2004*; *Galli et al., 2015*; *McSteen & Hake, 2001*), indicating the

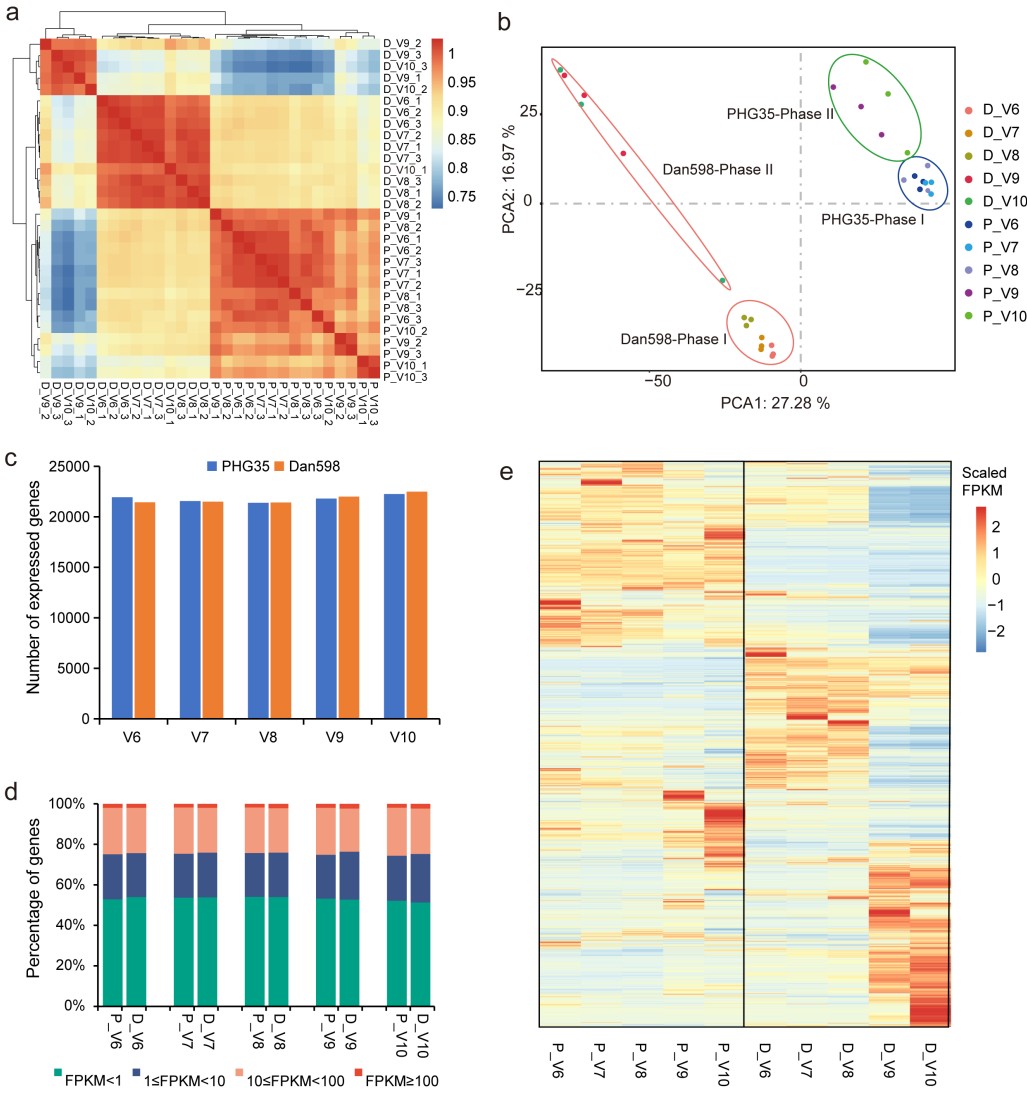

**Figure 2  Transcriptome profiling of early maize ear development in PHG35 and Dan598.** (A) Pearson correlation coefficient of the transcriptomes of the 30 young ear samples. (B) Principal component analysis (PCA) shows that similar samples or stages were clustered. The two lines were separated and two distinct developmental phases, Phase I (V6-V8) and Phase II (V9-V10), were present in both inbred lines. (C) Total number of genes expressed (FPKM ≥ 1). (D) Abundance distribution of detected genes (based on FPKM). The average FPKM value of the three replicates was calculated as the expression level for each stage in PHG35 and Dan598. (E) Expression profiles of all the expressed genes (FPKM ≥ 1) in PHG35 and Dan598 across the five stages of ear development. For each gene, the scaled FPKM values normalized using the z-score method were displayed by the continuous color scale. P, PHG35; D, Dan598; V6-V10, different development stages of young ears; 1-3 (in a), the three biological replicates.

significance of DEGs during the V6-V8 period for maize ears formation. Between V9 and V10, 2,543 common DEGs were specifically expressed, while a few genes controlling maize inflorescence architecture were identified, like *Zm00001d013603* (*KRN5b*, *Shen et al., 2024*).

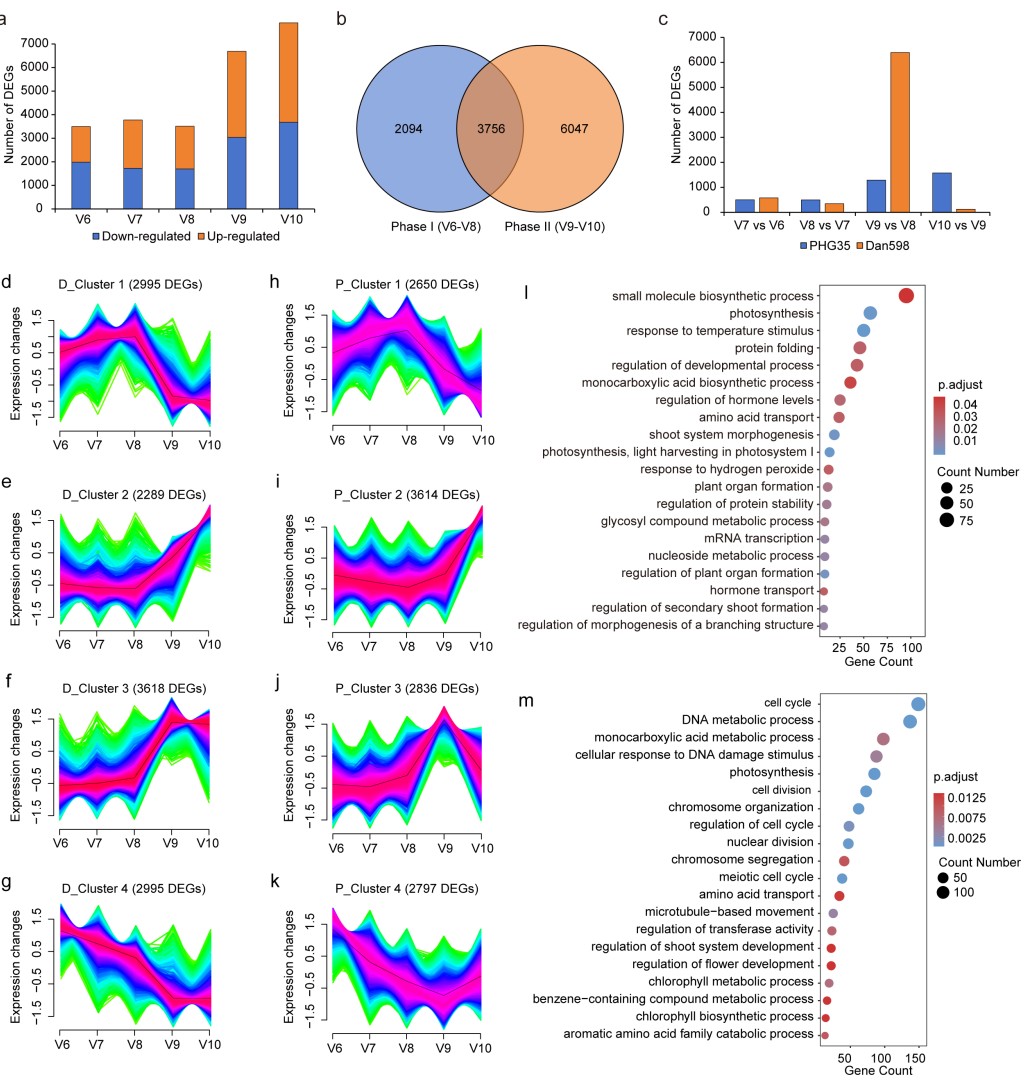

**Figure 3** **Differential gene expression in the two maize inbred lines at different stages of ear development.** (A) Screening of line-specific DEGs in inbred line Dan598 compared with PHG35 from V6 to V10. The orange segments represent the number of up-regulated genes and the blue segments down-regulated in Dan598. (B) Venn diagram displays the number of line-specific DEGs in Phase I and Phase II. (C) Screening of development-specific DEGs between adjacent stages in the two lines. (D–K) Fuzzy clustering of line-specific DEGs that were identified in Dan598 (D–G) and PHG35 (H–K) at five developmental stages using Mfuzz. Red indicates that genetic variations are most concentrated in the center of the cluster, followed by blue, and lastly green. The black line represents the expression trend of the cluster. (L) Top 20 biological processes in Phase I revealed by GO enrichment analysis. (M) Top 20 biological processes in Phase II revealed by GO enrichment analysis. The color indicates the p-adjust value, and the size of a dot indicates the number of genes detected.

To detect changes in gene expression levels at different ear development stages, we compared consecutive stages in each inbred line to identify development-specific DEGs. In PHG35, the expression of 503 genes (DEGs) significantly changed from V6 to V7, 499 DEGs were found from V7 to V8, and the DEG numbers became 1,291 and 1,576 from

V8 to V9 and from V9 to V10, respectively (Fig. 3C and Data S3). The pattern of DEG occurrence in Dan598 was seemingly different, with 580 DEGs between V6 and V7, 346 between V7 and V8, 6,394 between V8 and V9, and 115 between V9 and V10 (Fig. 3C and Data S3). These results showed that the number of development-specific DEGs in Dan598 was much larger than in PHG35, particularly at the period from V8 to V9.

## Expression patterns of DEGs during the ear development in the two inbred lines

As line-specific DEGs may better explain the variation in KRN between the two inbred lines, we performed Mfuzz clustering analysis on the 11,897 line-specific DEGs and identified four clusters in both Dan598 and PHG35. These clusters displayed diverse gene expression patterns (Figs. 3D–3K). Cluster 1 comprised 2,995 and 2,650 DEGs in Dan598 and PHG35, respectively, with expression levels gradually increasing from V6 to V8, peaking at V8, and then decreasing at V9 (Figs. 3D and 3H). Cluster 2 contained 2,289 and 3,614 DEGs in Dan598 and PHG35, respectively. The expression patterns showed low expression from V6 to V9 and then a significant increase at V10 (Figs. 3E and 3I). The expression patterns of the 3,618 DEGs in D_cluster 3 and the 2,836 DEGs in P_cluster 3 were lower from V6 to V8, showed an obvious increasing trend at V9, and then decreased at V10 (Figs. 3F and 3J). The expression level of the 2,995 DEGs in D_cluster 4 and the 2,797 DEGs in P_cluster 4 decreased gradually from V6 to V9 and slightly increased at V10 (Figs. 3G and 3K). Across the four clusters, V6 to V8 generally displayed different gene expression patterns compared to V9 and V10, suggesting an important developmental transition from V8 to V9, consistent with PCA clustering results. Previous studies have determined the vegetative growth stages by the leaf collar method (Abendroth et al., 2011; Ritchie, Hanway & Benson, 1986). Stages V6 to V8 begin 4 to 6 weeks after corn seedlings emerge from the soil. This phase is characterized by rapid growth and stem elongation, with the KRN being determined. Stages V9 and V10 mark the transition to a steady and rapid period of growth and dry matter accumulation. Therefore, we divided the five developmental stages into two distinct developmental phases: Phase I (V6 to V8) and Phase II (V9 and V10) in both PHG35 and Dan598.

We sought to examine whether there was a difference in GO terms between Phase I (V6 to V8) and Phase II (V9 and V10). Phase I contained 5,850 line-specific DEGs, and Gene ontology (GO) term enrichment analysis showed that these genes were significantly enriched in morphogenesis and differentiation processes, such as "regulation of plant organ formation", "regulation of morphogenesis of a branching structure", "plant organ formation", "regulation of hormone levels" and "hormone transport" (Fig. 3L). Previous studies have indicated that the majority of genes affecting KRN are highly expressed in the early stage of phase I and are enriched in the aforementioned biological processes (Shen et al., 2023). The 9,803 DEGs between the two inbred lines in Phase II (from V9 to V10) were significantly enriched in growth-related terms like "cell cycle", "DNA metabolic process", "chromosome organization", "nuclear division" and "meiotic cell cycle" (Fig. 3M). These findings indicate that Phase II represents a transition to rapid cell growth and cell proliferation, which corresponds to a key feature of ear development in this phase: the

rapid expansion of ear coincides with the differentiation of floral organs (*Abendroth et al., 2011*; *Ritchie, Hanway & Benson, 1986*).

## Mapping the line-specific DEGs in Phase I to KRN-related QTLs and QTN regions; prediction of candidate genes underlying known QTLs and QTNs

Considering the results of PCA clustering and GO term enrichment, we postulated that line-specific DEGs in Phase I are more likely involved in KRN determination in maize. Therefore, we mapped the 5,850 line-specific DEGs in Phase I to previously published KRN-related QTLs and QTN regions to further identify key regulatory genes for the KRN trait. A summary of KRN-related QTLs was shown in Data S4. LOD Score values of these QTLs range from 2.6 to 27.6, and PVE (%) range from 1.51 to 28.43. A total of 86 QTLs were identified across all ten chromosomes, and we depicted the 17 QTL hotspots (QTLs detected at least twice in previous studies) in Fig. 4, among which *qKRN4.2*, *qKRN5.1*, and *qKRN2.1* had been detected 21, eight, and seven times, respectively; *qKRN1.1*, *qKRN6.1*, and *qKRN10.1* were detected five times, and *qKRN4.1*, *qKRN5.4*, and *qKRN7.1* were detected four times. A total of 1,999 line-specific DEGs in Phase I were included in the 17 QTL hotspots. Among these DEGs, several genes are known to affect ear development. For example, *Terminal ear1* (*te1*, *Zm00001d042445*) was detected in *qKRN3.1* and is considered a marker of ear initiation (*White & Doebley, 1999*); *Zea floricaula/leafy1* (*zfl1*) and *zfl2* were detected in *qKRN10.1* and *qKRN2.1*, respectively. The *zfl1 zfl2* double mutant led to a disruption of floral organ identity and patterning, and the activities of these two genes are significantly associated with KRN (*Bomblies et al., 2003*). Meanwhile, 347 QTNs significantly associated with KRN were identified from previously reported studies (Fig. 4 and Data S4). Within 100 kb upstream and downstream of these associated QTNs, 225 genes were line-specific DEGs in Phase I. In summary, there were 2,132 DEGs identified in the 17 QTL hotspots and the 347 QTN flanking regions combined, among which 92 DEGs were within QTLs where QTNs also exist, thus they were stated to be within the common regions of QTL+QTNs (Fig. 4 and Data S5).

The 92 DEGs within the common regions of QTL+QTNs may contain candidate genes that have a high impact on KRN. Except for *qKRN2.2* and *qKRN8.1*, all the other QTL hotspots have a distribution of candidate genes (Fig. 4). Among them, *qKRN5.2* and *qKRN9.1* each contain one candidate gene, and *qKRN4.2* contains the highest number of DEGs (17 genes). The expression trends of these genes were shown in boxes next to the gene names. In addition, nine transcription factors (TFs) were identified in the common regions of QTL+QTN. For example, *Zm00001d002005*, encoding an SBP-box TF, was located within the QTL *qKRN2.1* with QTN snp55126 (Figs. 4 and 5B). *Zm00001d052365* (AP2/EREBP 155) was mapped in QTL *qKRN2.1* and QTN SYN3148, PZE-104109712 and SYN3156 regions (Figs. 4 and 5C). *Zm00001d054012*, encoding LRR-RLK, was located within QTL *qKRN4.2* and QTN snp43631 region (Figs. 4 and 5E). Two hormone signaling pathway genes were identified: *Zm00001d042267* (*ARF10*) was mapped to *qKRN3.1* with two significant QTNs (SYN5822 and PZE-103097885), and *Zm00001d025989* (*Aux/IAA*

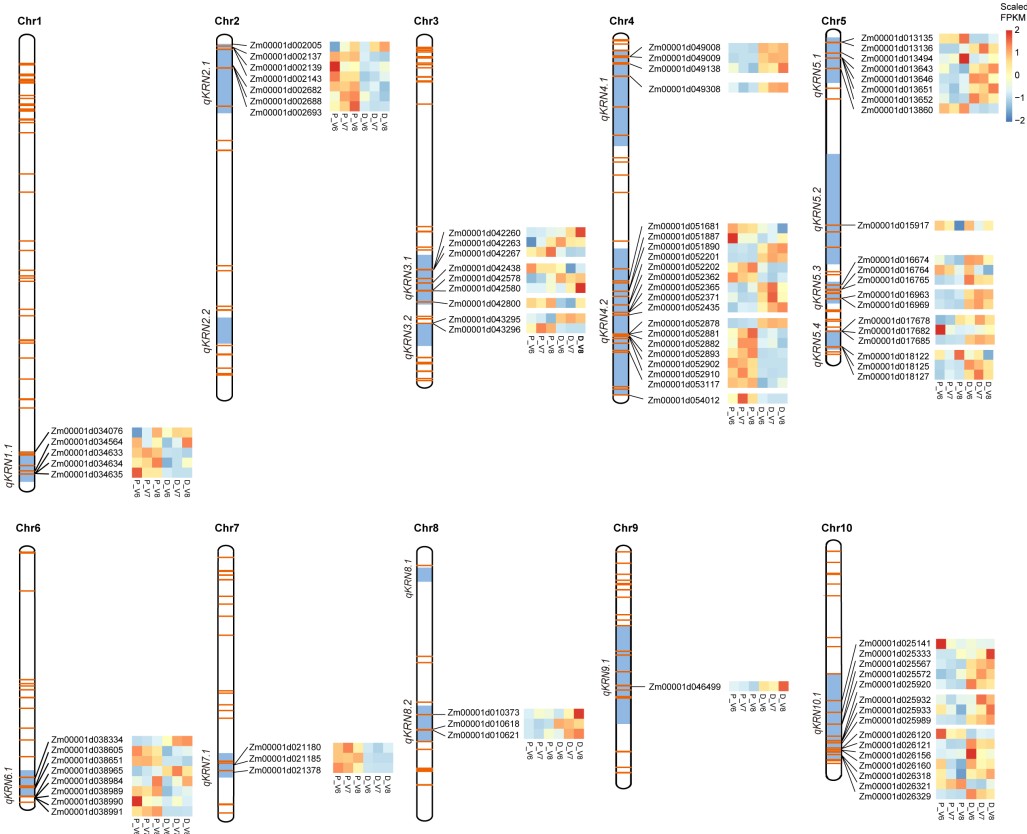

**Figure 4 Location of DEGs in KRN-related QTL hotspots and QTN regions.** QTL hotspots (QTLs that had been reported at least twice) were highlighted in blue boxes in all the ten chromosomes. The orange horizontal lines in the chromosomes represent known QTNs that are significantly associated with KRN. The genomic positions of the candidate genes in both QTLs and QTNs regions were displayed, and the DEG expression patterns in the two maize lines at V6 through V8 (Phase I) were shown in heatmaps. The scaled FPKM values were indicated by the continuous color scale. P, PHG35; D, Dan598.

*43*) was mapped to *qKRN10.1* with two significantly QTNs (PZE-110080302 and PZE-110080486) (Figs. 4 and 5F).

Among the 2,132 DEGs that fall into QTLs or QTN regions, a total of 168 TFs were identified, including numerous development-related TFs, such as SBP-box, AP2/ERF, and MADS-box TFs (Fig. 5A). We identified four SBP-box genes, all of which were significantly up-regulated in Dan598 compared with PHG35 (Fig. 5B). Fifteen AP2/EREBP TFs were identified, with ten genes were down-regulated and five genes were up-regulated in Dan598 (Fig. 5C). Four MADS-box TFs were identified and all were down-regulated in Dan598 (Fig. 5D). We also identified nine leucine-rich repeat receptor-like protein kinase family proteins (LRR-RLKs), among which five were down-regulated in Dan598 (Fig. 5E). In addition, many genes involved in hormone homeostasis and signaling, such as auxin and cytokinin, were identified (Figs. 5F and 5G). The auxin-activating enzyme *Zm00001d043701* (IAA-amino acid hydrolase ILR1-like 4) had higher gene expression in PHG35 at the V6 stage, while the auxin-inactivating gene *Zm00001d010697* (IAA-amide synthetase GH3.6)

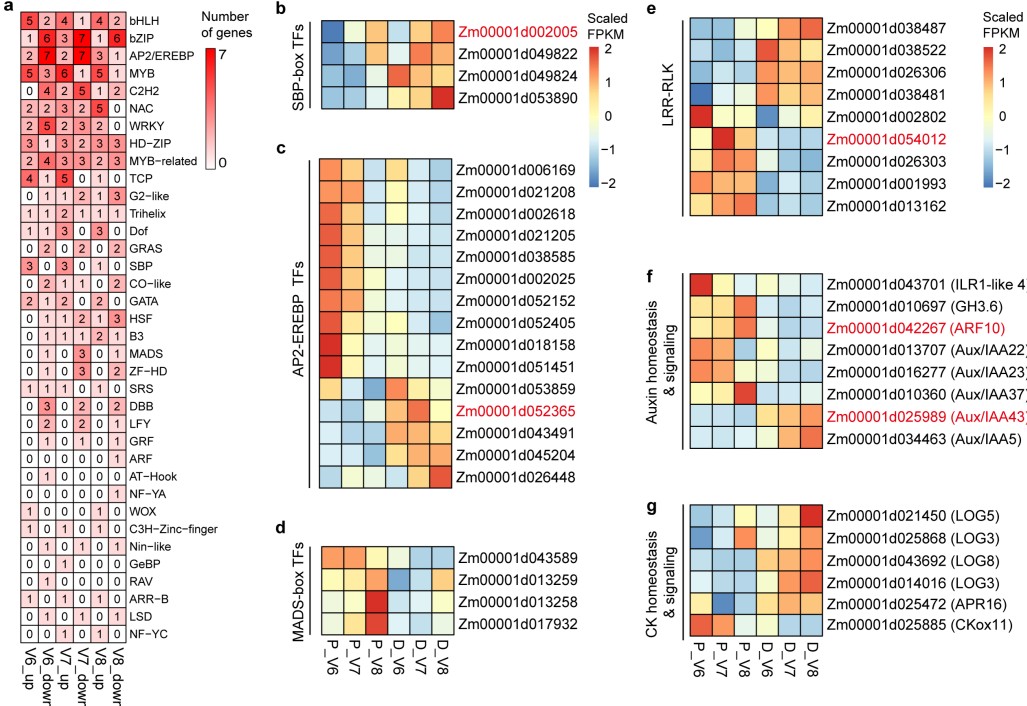

**Figure 5** **Differential expression pattern of transcription factors, LRR-RLKs, and hormone-related genes.** (A) Number of genes of different TF families among the 2132 candidate genes that are located in QTLs or QTN regions. Gene family names are listed at the right; the numbers in the boxes show the number of genes whose expression were detected to be up-regulated and down-regulated in Dan598 compared with PHG35 at each ear development stage (V6-V8). (B–G) The expression patterns in the two inbred lines of candidate genes in SBP-box TF family (B), AP2/EREBP TF family (C), MADS TF family (D), leucine-rich repeat receptor-like protein kinase family proteins (LRR-RLKs) (E), Auxin homeostasis and signaling pathways (F), and cytokinin homeostasis and signaling pathways (G). Genes in the common regions of QTL+QTN were displayed in red fonts. The scaled FPKM values were displayed by the continuous color scale.

had higher gene expression at the V8 stage in this line. Three auxin-responsive Aux/IAA family TFs (*Aux/IAA 22*, *Aux/IAA 23* and *Aux/IAA 37*) were down-regulated in Dan598, whereas, *Zm00001d034463* (*Aux/IAA5*) and *Zm00001d025989* (*Aux/IAA43*) were up-regulated in Dan598 (Fig. 5F). CK-activating enzymes (LOG3, LOG5 and LOG8) and CK signaling component TYPE-A RESPONSE REGULATOR 16 (ARR16) were up-regulated in Dan598. Meanwhile, CK inactivation gene *Zm00001d025885* (cytokinin oxidase11) was down-regulated in Dan598 compared with PHG35 (Fig. 5G).

## qRT-PCR verification

To vindicate our transcriptome data, we selected eight genes to verify gene expression by quantitative real-time PCR (qRT-PCR; Fig. 6). By comparison, qRT-PCR results matched the RNA-seq data reasonably well for all these genes, indicating that our RNA-seq results are reliable and can be used for elucidating the regulatory networks for ear development and KRN determination.

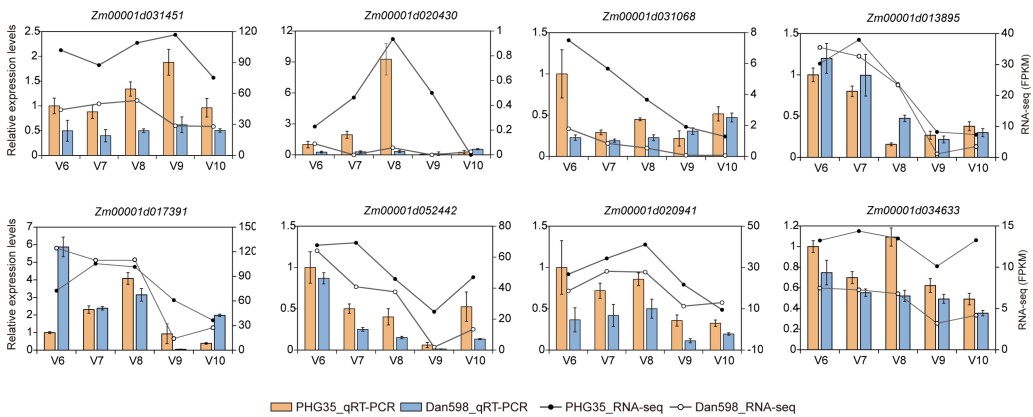

**Figure 6** **The expression levels of eight selected genes were measured using quantitative real-time PCR (qRT-PCR).** The bar graph depicts the qRT-PCR results. The expression level at V6 in PHG35 was set to 1. Data are presented as mean ± SD. The dashed lines show the RNA-seq data, and the FPKM values from RNA-seq are shown at the right axis. V6-V10 are the developmental stages.

## DISCUSSION

In this study, we utilized RNA-Seq methodology to detect the transcriptome dynamics in two maize inbred lines across five developmental stages of young ears (Table S2). Consistent with the vast phenotypic differences between the two maize inbred lines (PHG35 and Dan598), a total of 11,897 line-specific DEGs were detected between the two lines across the five development stages (from V6 to V10). Both PCA and Mfuzz clustering analysis showed a clear distinction from V8 to V9. Therefore, at the transcriptomic level, we divided the five developmental stages into two distinct developmental phases: Phase I (V6 to V8) and Phase II (V9 and V10) for both PHG35 and Dan598. Phase I contains 5,850 line-specific DEGs, which were significantly enriched in plant organ formation and hormone regulation. Line-specific DEGs in Phase II were significantly enriched in growth-related terms, indicating a transition to rapid ear growth and cell proliferation, which is distinct from the morphogenesis-dominated Phase I. As supported by previous studies, KRN is primarily determined during early developmental stages (*Abendroth et al., 2011*; *Ritchie, Hanway & Benson, 1986*), and gene expression levels at these stages play a critical role in controlling meristem fate and influencing KRN (*Shen et al., 2023*). Therefore, we propose that DEGs in Phase I (V6-V8) likely encompass the most promising candidate genes regulating KRN and warrant further investigation. In contrast, Phase II may contribute to final ear morphology by modulating ear expansion and floral organ differentiation.

At present, through QTL mapping and GWAS, numerous QTLs and QTNs associated with KRN trait in maize have been identified (*Dong et al., 2023*). Although hundreds of QTLs and QTNs have not yet been cloned, some QTLs and QTN hotspots at specific genomic regions have been detected in different populations multiple times (*Zhang et al., 2023*). Recently, RNA-Seq combined with QTL mapping and GWAS have been used to predict candidate genes underlying QTLs and QTNs associated with many traits, such as

iron content in maize (*Yan et al., 2023*), seed oil content in rapeseed (*Zhao et al., 2022*), and seed vigor in rice (*Guo et al., 2019*). In this study, a total of 17 QTL hotspots and 347 QTNs for KRN trait from previously reported studies were chosen as target loci for line-specific DEGs detected in Phase-I, and 2132 of the 5850 line-specific DEGs were mapped to these QTLs or QTN regions. Among these, 92 DEGs were located within the common regions of QTL+QTN. These results provided new clues for candidate genes underlying these QTLs and QTNs, which deserve further functional study for maize molecular breeding programs.

Transcription factors (TFs) bind to specific regulatory genomic regions to regulate the spatio-temporal expression of target genes. Several TFs have been identified that play vital roles in inflorescence development and KRN determination in maize. In our study, 168 TFs were differentially expressed and mapped to QTLs or QTN regions, including SBP-box, AP2/EREBP, LEAFY (LFY), and MADS-box TFs. In maize, *Unbranched3* (*UB2*) and *UB3* belong to SBP-box TFs, and mutants of *ub2* and *ub3* lead to an increase in KRN and enlargement of ear diameter (*Du et al., 2020*). AP2/EREBP TFs are involved in the development of IMs in maize. For instance, the gene *Indeterminate Spikelet1* (*IDS1*), encoding an AP2 domain protein, was cloned in the *qKRN1.1* locus controlling KRN (*Wang et al., 2019*). We identified four SBP-box and fifteen AP2 TFs. SBP-box TFs were significantly up-regulated in Dan598, whereas most AP2 TFs were down-regulated in Dan598 compared with PHG35. The results are consistent with that SBP-box TFs positively regulate miR172 expression, which in turn represses the AP2 TFs family (*Wang et al., 2021*). Additionally, The LFY TFs are meristem identity genes and are important in the promotion of cell proliferation and floral development (*Gao et al., 2019*). Here, two duplicated FLORICAULA/LEAFY homologs in maize, *zfl1* and *zfl2*, were detected in *qKRN10.1* and *qKRN2.1* regions, respectively. *zfl1* may be a potential candidate gene for *qKRN10.1*, which has been detected in five different studies and has not yet been cloned. However, further investigation is required to confirm it.

As small signaling molecules, plant hormones and peptides play various roles in regulating the development of inflorescence meristem. Auxin and cytokinins balance the proliferation and differentiation of meristematic cells through antagonistic effects (*Su, Liu & Zhang, 2011*). The reversible inactivation of auxin plays an important role in plant development mainly through the GH3-ILR1 pathway (*Hayashi et al., 2021*). Two enzymes implicated in auxin homeostasis, *Zm00001d010697* and *Zm00001d043701*, were located within *qKRN8.2* and *qKRN3.2*, respectively (*Fig. 5F*). In maize, *Barren inflorescence1* (*Bif1*) and *Bif4*, which are crucial for the initial stages of inflorescence development, encode proteins belonging to the Aux/IAA family. These proteins play a pivotal role in the auxin signaling pathway, a process that is indispensable for the formation of plant organs (*Galli et al., 2015*). *Bif4* were line-specific DEGs and were differently expressed from V6 to V10. We also identified five *Aux/IAAs* (*Aux/IAA5*, *Aux/IAA22*, *Aux/IAA23*, *Aux/IAA37*, and *Aux/IAA43*) that were differentially expressed between the two lines, with three down-regulated and two up-regulated in Dan598. Additionally, within the QTLs or QTN regions, we identified four cytokinin synthesis genes (LOG) and one cytokinin response gene (ARR16) that were up-regulated in Dan598, while one cytokinin oxidation gene was down-regulated in Dan598. Higher auxin metabolism was observed in PGH35, whereas

higher cytokinin synthesis and response were observed in Dan598. The results indicate significant differences in auxin and cytokinin levels between the two lines, suggesting that auxin and cytokinin may play opposite roles in regulating ear development. The CLAVATA-WUSCHEL (CLV-WUS) negative feedback loop, first identified in Arabidopsis, maintains a balance between stem cell proliferation and ongoing cellular differentiation for organ initiation (*Schoof et al., 2000*). *CLV1* and *CLV2* encode LRR-RLK protein and LRR receptor-like protein, respectively. In maize, *thick tassel dwarf1* (*td1*) and *fasciated ear2* (*fea2*) are orthologous to *CLV1* and *CLV2*, respectively, and regulate the size of the tassel and ear IM (*Bommert et al., 2005*; *Bommert, Nagasawa & Jackson, 2013*). In this study, a known KRN-related gene *fea3*, (*Je et al., 2016*), encoding an LRR receptor-like protein, was differentially expressed between the two lines in both the V6 and V7 stages. Moreover, nine LRR-RLKs were located in the QTLs or QTN regions, among which *Zm00001d054012* was located in the common region of QTL *qKRN4.2* and QTN snp43631. These results imply the important roles of the CLV-WUS pathway during maize ear development, which provides a basis for the subsequent improvement in KRN.

## CONCLUSIONS

Our high-resolution transcriptome analysis revealed that the five stages can be divided into two distinct phases: Phase I (V6-V8) and Phase II (V9-V10). A total of 11,897 line-specific DEGs were identified between the two inbred lines, of which 5,850 line-specific DEGs in Phase I are more likely involved in KRN determination. By integrating these DEGs with published KRN-related QTLs and QTNs, we identified 2,132 high-confidence candidate genes. Among these, 92 DEGs co-localized in overlapping QTL+QTN regions, including TFs, genes involved in plant hormones homeostasis and signaling, and peptides. Future functional validation of these high-confidence candidate genes through CRISPR-Cas9 editing or transgenic approaches will clarify their specific roles in KRN regulation. Our framework also offers a roadmap for dissecting other agronomic traits by coupling omics analyses with genetic architecture analysis.

### Funding

This work was supported by the National Natural Science Foundation of China to Cuixia Chen (No. 32370672), the National Key R&D Program of China to Cuixia Chen (2023YFD1200501). The funders had no role in study design, data collection and analysis, decision to publish, or preparation of the manuscript.

### Grant Disclosures

The following grant information was disclosed by the authors:
The National Natural Science Foundation of China: No. 32370672.
The National Key R&D Program of China: 2023YFD1200501.

## Competing Interests

The authors declare there are no competing interests.

## Author Contributions

- Shukai Wang conceived and designed the experiments, performed the experiments, analyzed the data, prepared figures and/or tables, authored or reviewed drafts of the article, and approved the final draft.
- Yancui Wang conceived and designed the experiments, performed the experiments, analyzed the data, prepared figures and/or tables, and approved the final draft.
- Xitong Xu performed the experiments, prepared figures and/or tables, and approved the final draft.
- Dusheng Lu performed the experiments, prepared figures and/or tables, and approved the final draft.
- Baokun Li performed the experiments, authored or reviewed drafts of the article, and approved the final draft.
- Yuxin Zhao performed the experiments, authored or reviewed drafts of the article, and approved the final draft.
- Senan Cheng performed the experiments, analyzed the data, authored or reviewed drafts of the article, and approved the final draft.
- Zhenhong Li performed the experiments, authored or reviewed drafts of the article, and approved the final draft.
- Cuixia Chen conceived and designed the experiments, analyzed the data, authored or reviewed drafts of the article, and approved the final draft.

## Data Availability

The RNA-seq raw sequence data are available in the Genome Sequence Archive: CRA020486.

The raw data is available in the Supplemental Files.

## Supplemental Information

Supplemental information for this article can be found online at http://dx.doi.org/10.7717/peerj.19143#supplemental-information.

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
