# Peer review of "Comparative transcriptome analysis identified candidate genes associated with kernel row number in maize"

_PeerJ, doi:10.7717/peerj.19143_

## Round 0.1 · original submission · Minor Revisions

After you make minor revisions in accordance with the reviewers' reports, your manuscript will be evaluated for acceptance.

Reviewer 1 ·

Basic reporting

You can find in attached pdf

Experimental design

-

Validity of the findings

-

Additional comments

-

Annotated reviews are not available for download in order to protect the identity of reviewers who chose to remain anonymous.

·

Basic reporting

Kernel row number is one of the important yielding attributing traits in maize. Basic genetic information on KRN is already known and application of basic knowledge in breeding is limited. This may be because of the lack of proper tools to integrate the genetic information available with the routine breeding program. In this connection, a transcriptome-based approach to understanding the novel gene and trying to use the same in crop improvement may give new insights into the traits and their utilization.

In this Introduction, the author has mentioned 4 QTLs have been cloned, but it appears still more QTLs are still cloned, that may be mentioned

Experimental design

Material methods were clear but the Genetic background of the genotypes and the sources from which it was derived, may be mentioned
Give details on library preparation in MM
Special expression of the gene is required if not it may be discussed

Validity of the findings

Results appear to be appropriate and validated
221 to 223, Is this from your result? or else justify with reference.

Additional comments

The manuscript is written meticulously. All the sections are well explained. However, there are some clarifications required in some sections, that may be attempted.

Reviewer 3 ·

Basic reporting

no comment

Experimental design

1. The PHG35 and Dan598 were planted in the experimental field of Shandong Agricultural University (Taian, China). The authors should introduce the planting density and conditions.

Validity of the findings

1. In this study, a total of 30 cDNA libraries were constructed for sequencing. However, a sample of D_V9 showed greater separation in Figure 2e, whether that has an impact on results?
2. The five developmental stages can be divided into two distinct developmental phases: Phase I (V6 to V8) and Phase II (V9 and V10) in both PHG35 and Dan598. The authors should introduce the basis in detail.
3. In this study, the authors obtained 8612 line-specific DEGs in Phase II (V9-V10). I think that it is important to understand the molecular mechanism of KRN development. The authors should add the Phase II analysis.
4. The discussion section should be revised in order to better understand the results, rather than “Their similar expression patterns suggest that they may have conserved functions.”

---

## Round 0.2 · Minor Revisions

Your manuscript needs some final very minor revisions.

Reviewer 1 ·

Basic reporting

From my side it is ok. The author did the required editing

Experimental design

-

Validity of the findings

-

Additional comments

-

Reviewer 3 ·

Basic reporting

The author has greatly improved the manuscript, however, the authors should revise their conclusion and clearly summarize their findings.

It is recommended to add a description of the Phase Ⅱ.

Experimental design

no comment

Validity of the findings

no comment

Additional comments

no comment

---

## Round 0.3 · accepted · Accept

Your manuscript accepted after last revision. Congratulations